# International Capital Flows and Speculation

Rob Hayward *,†  and Andros Gregoriou †

School of Business and Law, University of Brighton, Brighton BN2 4AT, UK; andros.gregoriou@brighton.ac.uk
* Correspondence: rh49@brighton.ac.uk; Tel.: +44-1273-642-584
† These authors contributed equally to this work.

**Abstract:** In response to questions about the relative importance of different types of capital flow for international competitiveness, we develop a structural vector auto-regressive model of the real exchange rate and international capital flows. We reveal that innovations to speculative sentiment cause changes in competitiveness. We report that speculation replaces the effect of equity, bond and most of the interest rate effect. The results show that international speculative sentiment is an important contributor to exchange rate and that monetary and regulatory authorities should find ways of measuring and understanding banking and financial flows.

**Keywords:** exchange rates; currency crisis; capital flow; VAR



## 1. Introduction

The rise in gross and net international capital flows that has taken place in the last 20 years has been documented by (Lane and Milesi-Ferretti 2007), (Obstfeld and Taylor 2004), the (BIS 2013) and others. The amount of foreign exchange that is traded by other financial institutions reached a daily average of USD 3.6 trillion in April 2019 according to the BIS Triennial Survey of Turnover (BIS 2019). While the free flow of global capital should allow smoothing of consumption, sharing of risk and the financing of global projects that have the greatest return, evidence has accumulated that the link between international capital flows and economic development is not as prominent in practice as it appears to be in theory.

Indeed, the study of international financial crisis has shown that speculative flows of capital can cause major financial disruption, particularly where countries experience sudden stops to the inflow of funds, (Calvo 1998), (Dornbusch and Werner 1995) and (Krugman 2000). A key finding of this research is that not all capital flows are the equivalent: some are contractionary and some are expansionary (Blanchard et al. 2015); some can be swiftly reversed while others are more sticky and difficult to withdraw (Chuhan et al. 1996) and (Claessens et al. 1995). (Keefe 2021) report that international interest rate and credit shocks propagated through capital flows have become more pronounced since the Global Financial Crisis (GFC) in 2008–2009. Therefore, measuring and understanding the cause, determination and nature of these shocks is of great interest to policymakers and researchers.

Exchange rate models that incorporate capital flows have been developed. However, the process has been impeded by the availability of data and by the difficulties of modelling the interaction between capital flows. Cross-border data about Foreign Direct Investment (FDI) and the purchase of debt and equity are available but understanding and untangling the adjustments in banking assets and deposits is more challenging. This paper contributes to our understanding of international flow of capital by developing a structural vector auto-regression (SVAR) framework to model capital flows and the real exchange rate, adding speculation to the capital flow model by using a unique measurement from the Commodity Futures Trading Commission (CFTC) Commitment of Traders Report (COT) to identify the effect of speculative flow of capital on competitiveness.

Impulse response functions (IRF) showing how innovations to capital flows can affect the real exchange rate are presented. A one standard deviation innovation or shock to speculative sentiment leads on average to a 2 percentage point increase in the US real trade weighted index. There is also momentum behind the changes in the real exchange rate. A one standard deviation innovation or shock to the real exchange rate tends to be followed by a 2 to 3 percent adjustment in the same direction of that shock. The influence of other capital flows on the exchange rate is at best ambiguous. The effect of bond, equity and foreign direct investment shocks are economically and statistically insignificant once speculative flows are accounted for; interest rate differential have a small positive effect. Bond purchases by central banks are associated with a weaker Real Trade Weighted Index (RTWI) in a what is likely to be a central bank response to overseas currency weakness. The results are resilient to alternative model specifications.

The rest of this paper is organised as follows: Section 2 reviews the literature; Section 3 covers methods and data; Section 4 presents the results; Section 5 concludes.

## 2. Literature Review

Understanding the international flow of capital is a key interest of policymakers and the research community. The effect of capital flow on exchanges rates includes excess volatility and changes in the real exchange rate that can influence economic growth, the price level as well as asset prices and policy options. This understanding relies on availability of clear capital flow data. However, while there is information about the international purchase of stocks and bonds as well as the demand for Foreign Direct Investment (FDI), the more short-term and speculative reasons for purchasing foreign currency are hidden within the international banking statistics. One of the core findings of the 2011 report from the Committee for International Economic Policy and Reform, Rethinking Central Banking, was the need for more adequate data to keep track of the complex matrix of gross cross-border capital flows and the gross external asset and liability positions of countries (Prasad et al. 2011).

Theoretical models have traditionally been based on the net capital flows. For example, (Gabaix and Maggiori 2015) examine exchange rates by assessing the way that net capital flows affect the balance sheets and risk appetite of financial institutions. Changes in risk appetite affect the volatility and level of exchange rates. (Devereux et al. 2020) use net capital flows to understand Net International Investment in the US and other countries. World wealth is determined by changes in bond risk premia. Hau and Rey present a model that links exchange rates, stock prices and net capital flows. Using data from the transactions of US global mutual funds they find that higher returns in the home equity market relative to overseas are associated with domestic currency depreciation while net equity flows into a foreign equity market are associated with a foreign currency appreciation (Hau and Rey 2006). The importance of understanding gross flows of capital has become increasingly prevalent. (Caballero and Simsek 2020) show how gross capital flows can explain fickleness of foreign investments when there are shocks and fire-sales of financial assets. Thereby gross flows of capital can encourage policymakers to impose capital controls to prevent the search for safety or yield from destabilising their economy.

However, all these models concentrate on international purchase of equity and debt. The structural model of international capital flows and the real exchange rate has a downward sloping demand curve. For example, the net purchase of US equities by overseas investor will tend to increase the value of the US dollar (Hau and Rey 2006), though financial or natural hedging may limit the effect in some cases (Bathia et al. 2020). In all these models there must be an offsetting money market flow that finances the asset purchase. The models treat money market transactions between countries as the facilitating payment for goods, services and international financial assets. They cannot capture the deliberate effect of money market flows that may be determined by exchange rate or interest rate expectations, that are not directly linked to purchase of equities, bonds or real assets (Kumhof et al. 2020).

Therefore, looking at either the net or gross flow of portfolio assets gives an incomplete picture of the exchange rate pressure caused by international investment. (Borio and Disyatat 2011) criticise the narrative that a savings glut led to global imbalances and contributed to the GFC by highlighting the way that stories about current account imbalances fail to discuss gross financial flows and their contributions to changes in the stock of assets or the role that these flows play in international borrowing, lending and financial intermediation. They show that cross-border bank lending and bank deposits account for a very large share of major economies' overall cross-border gross positions. They argue that it is excess elasticity of financing rather than a savings glut that contributed to the GFC.

(Cesa-Bianchi et al. 2019) present a model where banking assets and liabilities determine exchange rates with bank loans as an enhanced theory of the monetary theory of exchange rate determination where broad money is determined by commercial banks. Here the loan and deposit creation is designed for the transaction of goods and services but it could equally well be for the purchase and sale of financial assets. They show that shifts in investors sentiment cause shocks to money demand and exchange rates. The belief that more attention should be paid to international banking than to the purchase and sale of equities and bonds is taken up by (Lane and Milesi-Ferretti 2008) when they discuss the way that financial integration has driven financial globalisation. (Obstfeld 2010) highlights the way that the curtailment of dollar funding for European commercial banks after 2008 caused upward pressure on the US dollar. Here it is the change in liquidity and credit that are affecting exchange rates rather than the purchase of debt or equity.

Lack of access to detailed data can confute understanding of key mechanisms in international finance. (Avdjiev and Zeng 2014) construct data on sectoral gross capital flows as a response to the variety of underlying causes of international financial shocks. This is a quarterly panel of 31 countries. It is broken down into public sector, private sector and banking sector but aggregated across assets. They report the importance pro-cyclical outflows of capital from Advanced Economy banks and the stabilising role of the Emerging Economy public sector. They also report they difficulties of constructing their data as well as the rather limited resources that are available from institutions like the EPFR and IFF.

## 3. Methods and Data

There are three major methodological issues that have to be overcome when assessing the effect of capital flows on exchange rates: The equilibrium exchange rate, the variables that will be included in the model and the issue of endogeneity. The task here is to model fluctuations in the real trade-weighted US exchange rate using flows of international capital to describe changes in competitiveness with the aim of identifying the relative importance of different types of capital flow. The US is chosen as it has the most detailed data for portfolio flows and speculative positions. The real trade-weighted US exchange rate is allows us to say something about international competitiveness.

### 3.1. The Data

As identified in the review of literature, identifying and measuring speculative flows has been a challenge due to the scarcity of data. A standard approach has been to utilise (BIS 2012). These figures record cross-border banking exposure. For example, (Bruno and Shin 2014) model the global liquidity cycle of international banks; (Adams-Kane et al. 2015) try to assess the relative importance or willingness to lend, the level of liquidity and measures of solvency in determining capital flow. However, it is impossible, using this data, to distinguish between bank lending for real business projects and that for speculation in financial markets. As a rare exception, (Ceruttie et al. 2014) combine BIS with proprietary banking data to analyse the speculative flow between international and local banks.

International demand for US bonds and equities is measured here using data from the US Treasury. Since January 1977, the US Treasury has released a monthly report providing significant detail about the changes in the holding of long-term securities amongst US and overseas investors. This report is part of a series of reports under the Treasury International

Capital Department (commonly known as the TIC data). Gross inflows and outflows are netted, summed over time and normalised to GDP to create a variable that is equivalent to the net stock of US equity and debt relative to GDP. The US Treasury also release data about the investments of international monetary authorities. These can be used to give an indication of the foreign exchange intervention of central banks. See (Siourounis 2004) and US Treasury (US Treasury 2014) for a comprehensive overview of this data series. A measure of US net FDI assets relative to GDP is created in similar fashion using data from the US Bureau of Economic Analysis on International Transactions (US Department of Commerce 2014).

Measuring short-term, speculative money-market flows is more challenging. There are two ways that money market speculation is assessed here. The first series are compiled to account for for associated with sentiment, momentum or technical trading. The series for these flows are the positions held by speculative funds in the main currency futures markets in the US. These are positions that must be reported to the US derivative regulator, the US Commodity Futures Trading Commission (CFTC). The positions are held in foreign currency vs. the US dollar. The key contracts are Canadian Dollar (contract of 100,000 Canadian dollars), UK Sterling (contract of 62,500 sterling), Japanese yen (12,500,000 yen), Swiss franc (125,000 CHF) and Euro (125,000 EUR) or Deutschmark before the introduction of ECU trading. The data and explanation about the differentiation between commercial (hedgers) and non-commercial (speculators) is available from the CFTC web site. The outstanding long or short speculative positions are amalgamated across currencies and normalised to the total number of speculative positions or the total open interest positions to get an overall measure of sentiment. The interest rate spread between the US and the rest of the world is also used to capture short-term interest rate flows such as the carry-trade. Additional information about the measurement of speculative flows is available in (Hayward 2018).

Therefore, there are seven variables that are to be used in the analysis and a number of variations that can be applied to the model. The primary model is reported here. The main variables are: the cumulative net bond per GDP ($CNB$); the cumulative net equity per GDP ($CNE$); cumulative net foreign direct investment per GDP ($CNFDI$); cumulative net official treasuries per GDP ($COT$); the real trade-weighted index ($RTWI$); the spread between US short rates and the rates of the main trading partners ($SPREAD2$); and, a measure of speculative sentiment ($S1$). The full data set run from the first quarter of 1973 through to the first quarter of 2020. Figure 1 shows the evolution of the main variables used in the study during the study period.

### 3.2. Identification

Statistical difficulties emerge when trying to estimate parameters when there is feedback from the dependent variable onto explanatory variable. (Kouri and Porter 1974) show that a linear regression of capital flows on interest rate differentials will systematically under-estimate the sensitivity of capital flows. The (Brooks et al. 2004) model suffers in the same way from being reduced form. Vector Auto Regression (VAR) is one way of dealing with the issue of endogeneity. The method was initially suggested by (Sims 1980), An overview of developments and extensions can be found in (Lutkephole 2006) and (Hamilton 1994). The essence of the VAR is to create a system with all the important variables, assume that they are endogenous and add significant lags to remove any serial correlation from the residuals.

**Cumulative capital flow and exchange rate**

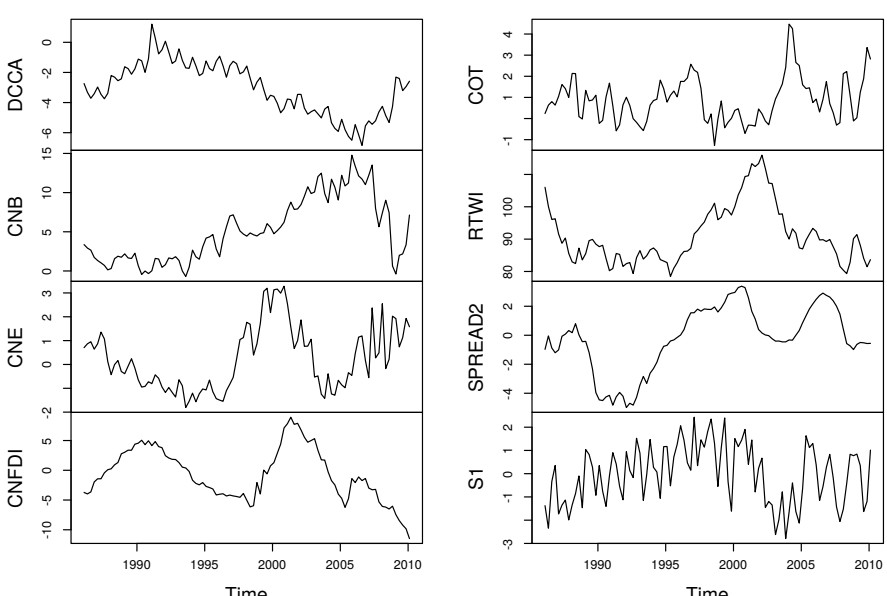

**Figure 1.** Cumulative capital flows and USD.

A simple structural or primitive system with just two endogenous variables and one lag would be.

$$x_t = b_{10} + b_{12}y_t + \gamma_{11}y_{t-1} + \gamma_{12}x_{t-1} + \epsilon_{xt}, \tag{1a}$$

$$y_t = b_{20} + b_{22}x_t + \gamma_{21}x_{t-1} + \gamma_{22}y_{t-1} + \epsilon_{yt}, \tag{1b}$$

It is assumed that $\epsilon_{xt}$ and $\epsilon_{yt}$ are independent of each other, are normally distributed with a constant variance. Though the independence assumption appears quite onerous, it should be remembered that this is independence of innovation. It does not mean that there is no relationship between the two variables ($x$ and $y$ here). A shock to either variable will have an effect on the other via the contemporaneous relationship (which is $b_{11}$ and $b_{21}$ in Equation (1a) and (1b)). As a consequence, this model cannot be easily estimated as there are feedback effects of the sort discussed above ($y_t$ is related to $\epsilon_{xt}$ and $x_t$ is related to $\epsilon_{yt}$). The variables are endogenous. As an alternative to the use of Instrumental Variables the structural or primitive equations can be transformed so that ordinary least squares (OLS) can be utilised.

In matrix form, (1a) and (1b) can be written as,

$$\begin{bmatrix} 1 & b_{12} \\ b_{21} & 1 \end{bmatrix} \begin{bmatrix} x_t \\ y_t \end{bmatrix} = \begin{bmatrix} b_{10} \\ b_{20} \end{bmatrix} + \begin{bmatrix} \gamma_{11} & \gamma_{12} \\ \gamma_{21} & \gamma_{22} \end{bmatrix} \begin{bmatrix} x_{t-1} \\ y_{t-1} \end{bmatrix} + \begin{bmatrix} \epsilon_{xt} \\ \epsilon_{yt} \end{bmatrix} \tag{2}$$

A general version of the system can be represented as

$$Bx_t = \Gamma_0 + \Gamma_1 x_{t-1} + \epsilon_t \tag{3}$$

where $B$ identifies the reduced form (to be discussed in more detail below), $x_t$ is a vector of variables in the system (real exchange rate and capital), $\Gamma_0$ and $\Gamma_1$ are the coefficients to be estimated and $\varepsilon_t$ is a matrix of residuals. A key requirement of the method is that the residuals are well behaved. Tests for autocorrelation and hetroscedasticity of the residuals are discussed below.

If we multiplying through by $B^{-1}$ we get the standard form of the VAR

$$x_t = A_0 + A_1 x_{t-1} + e_t \tag{4}$$

with $A_0 = B^{-1}\Gamma_0$, $A_1 = B^{-1}\Gamma_1$ and $e_t = B^{-1}\epsilon_t$, and returning to the equivalent form

$$y_t = a_{10} + a_{11}y_{t-1} + a_{12}x_{t-1} + e_{1t} \tag{5}$$

$$x_t = a_{20} + a_{21}x_{t-1} + a_{22}y_{t-1} + e_{1t} \tag{6}$$

Now the equations can be estimated with OLS because the variables are uncorrelated with the error term. However, to understand the contemporaneous effect of a change in one variable on another the model needs to be identified so that unknown coefficients are equal to the number of equations in the VAR model. The Sims–Bernanke approach is adopted here with specific restrictions imposed (Sims 1986) and (Bernanke 1986). However, as a robustness check the model is also identified using the Cholesky decomposition. The results are broadly the same under each specification. There are are seven variables in the model being estimated. This means that twenty one restrictions ($K(K-1)/2$) are needed for system identification.

Twenty one restrictions are placed on the model using plausible economics assumptions. Table 1 gives an overview of of the restrictions suggested. These are the restrictions placed on the B matrix in Equation (3).

The following are the explanations for the twenty-one restrictions that are imposed on the VAR. Note that these are contemporaneous restrictions (same quarter), a lagged effect is still allowed. The equations are read across rows, with the dependent variable normalised to unity, coefficients on independent variables that are to be estimated are labelled "NA" in Table 1.

**Table 1.** The table shows the individual equations in the VAR and the restrictions that are placed on some of the coefficients to identify the system.

|        | CNB | CNE | CNFDI | COT | RTWI | SPREAD | S1 |
|--------|-----|-----|-------|-----|------|--------|-----|
| CNB    | 1   | NA  | 0a    | 0b  | 0c   | NA     | 0d  |
| CNE    | NA  | 1   | NA    | 0e  | NA   | 0f     | NA  |
| CNFDI  | g   | NA  | 1     | 0h  | NA   | 0i     | 0j  |
| COT    | NA  | 0k  | 0l    | 1   | NA   | 0m     | NA  |
| RTWI   | 0n  | NA  | NA    | NA  | 1    | NA     | NA  |
| SPREAD | NA  | 0p  | 0q    | 0r  | NA   | 1      | 0s  |
| S1     | 0t  | NA  | 0u    | NA  | NA   | 0v     | 1   |

Reading across the row for each equation in turn. The cumulative net bond Equation (CNB) is restricted by imposing a coefficient of zero on the influence of foreign direct investment (a), cumulative official treasuries (b), real exchange rate (c) and sentiment (d). Though the exchange rate and speculative sentiment could increase net bond flows, it seems more likely that this would happen at the short end of the yield curve (and therefore would be better captured by the money market proxies SPREAD or S1) and, as noted in (Siourounis 2004, p. 3) and (Hau and Rey 2006), most of the international bond flows appear to be hedged against foreign exchange gains and losses. The cumulative net equity equation is restricted only at the cumulative official treasuries (e) and the interest rate spread (f). Lower relative rates could inspire a more positive attitude towards corporate profits, but rate changes could just as likely be a response to broad-based economic weakness that would not be conducive to profitability. The restrictions on the FDI equations are on bond flows (g), official treasuries (h), the interest rate spread (i) and sentiment (j). As foreign direct investment is assumed to be a more long-term commitment, it seems likely that short-term relationship with other variables will be modest; the longer term coefficients can play a more prominent role. The flow of Official Treasuries (COT) is most likely to be a response to an appreciation of the US dollar and therefore should not be significantly affected by things like equity (k) and FDI (l) flows, unless indirectly. The real exchange rate is allowed to be affected by all the other variables outside of net bond flows (n). The interest-rate spread, which presumably is largely a function of central bank policy, is restricted against

net equity (p), net FDI (q), official purchase (r) and sentiment (s). The exchange rate and net bond flows are allowed to have some influence. Finally, the sentiment equation is restricted on bonds (t), FDI (u) and the spread (v), but is allowed to be affected by equity and the exchange rate. This allowed for some positive spillover from more optimistic attitudes towards the economy, which may affect the flow of money to the stock market or into real investments. It also allows for positive feedback from a change in the value of the exchange rate to speculative sentiment.

As a robustness check, the VAR is also run with a Cholesky decomposition which forces the error variance-covariance matrix to become an upper triangle and therefore imposes the required $K(K - 1)/2$ restrictions. This is a ad hoc or naive method that make the ordering of the variables important for the restrictions. However, it does allow a comparison of the model under two method of restriction. This allows the importance of the imposed restrictions to be assessed.

## 4. Results

There are a number of different models tested. The alternatives include the use of the current account to GDP ratio as an explanatory variable, a number of dummy variables and different ways of measuring the interest rate spread and speculation. The preferred model, according to Information Criteria and diagnostic statistics is the one that does not include the current account, has four lags and uses the three dummy variables as well as the SPREAD2 and S1 measures of interest rate and speculation. The model also has a linear trend and a constant. It has the lowest readings for Akaike Information (AIK) and Bayesian Information (BIC) Criteria. The system is stable as the roots of the autoregressive coefficients are all within the unit circle. The dummy variables are significant.

Impulse response functions (IRF) showing how innovations to capital flows can affect the real exchange rate when a set of considered restrictions are presented in Figure 2. These are the graphical representations of the effect of a one unit shock or innovation on one part of the system to any other part of the system. The IRF show how the system deals with disturbances. Technically, this is achieved by using the moving average (MA) version of the VAR and then applying some restrictions to ensure that the shocks can be identified. A one standard deviation innovation or shock to speculative sentiment leads on average to a 2 percentage point increase in the US real trade weighted index. There is also momentum behind the changes in the real exchange rate. A one standard deviation innovation or shock to the real exchange rate tends to be followed by a 2 to 3 percent adjustment in the same direction of the shock. The influence of other capital flows on the exchange rate is more ambiguous. The effect of bond, equity and foreign direct investment shocks are economically and statistically insignificant; interest rate differential appear to have a small positive effect. Bond purchases by central banks, presumably tied to official interventions are associated with a weaker Real Trade Weighted Index (RTWI) in a what is likely to be a reversal of causation.

As a robustness check, alternative ways of identifying the model are also assessed. The first of these will force the B matrix to be a lower triangle based on the initial order of the endogenous variables (the order is that of Table 1); the second is the same but the order of the variables is selected randomly. If the three different systems produce three different results then it can be concluded that the method used for identification plays a large part in the results. This would raise doubts about the reliability or robustness of the results. However, the results are very similar.

As Figure 3 shows the results do not depend on the restrictions that are applied to the SVAR. Using a parsimonious Cholesky decomposition to identify the system produces the same fundamental finding that speculative sentiment drives that real exchange rate while conventional capital flows do not. This finding is also robust to alternative measures of interest rates and the use of dummies for the sharp increase in US interest rates in 1994 and during the global financial crisis.

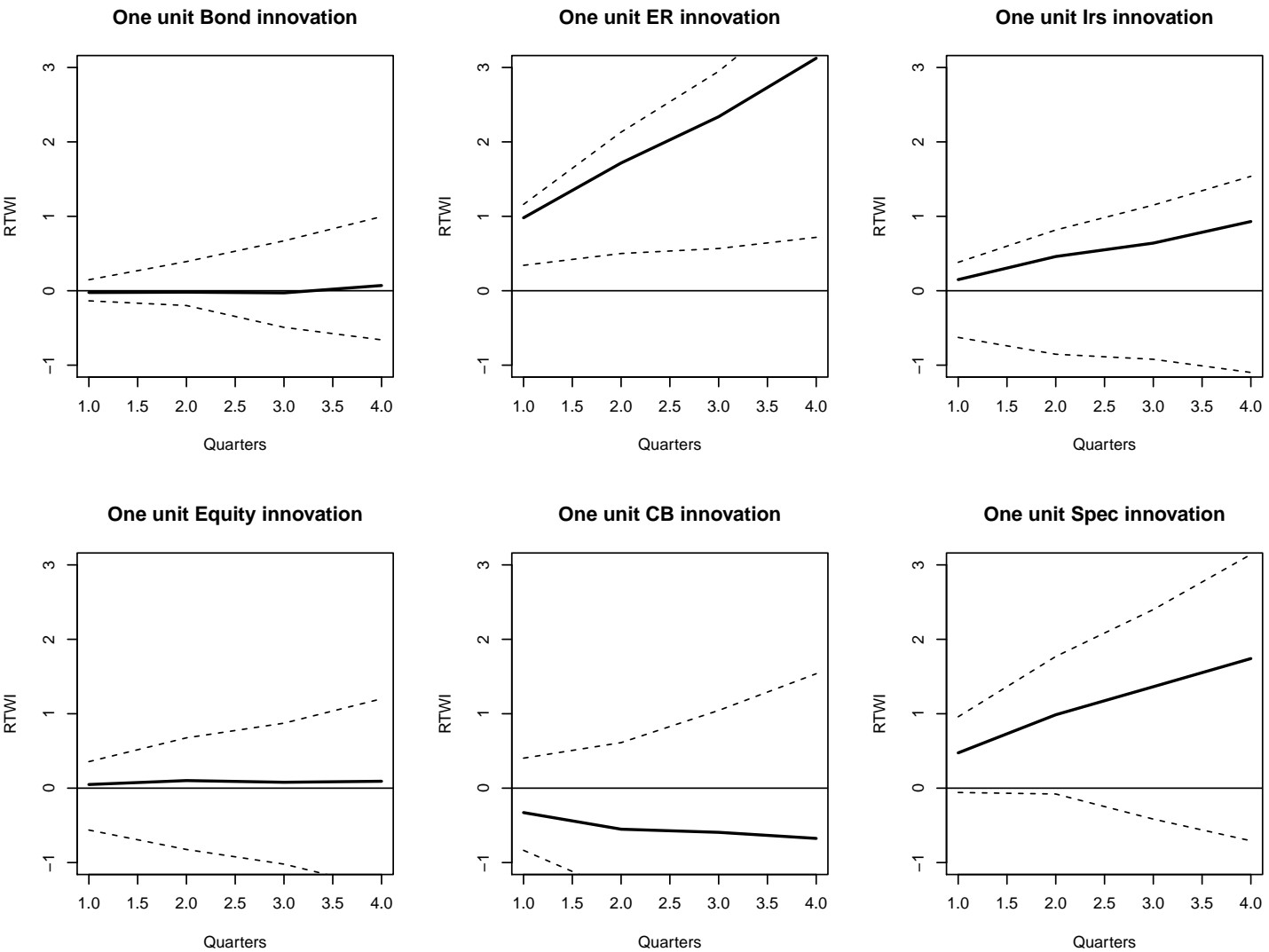

**Figure 2.** Impulse Response Functions for RTWI: SVAR identification.

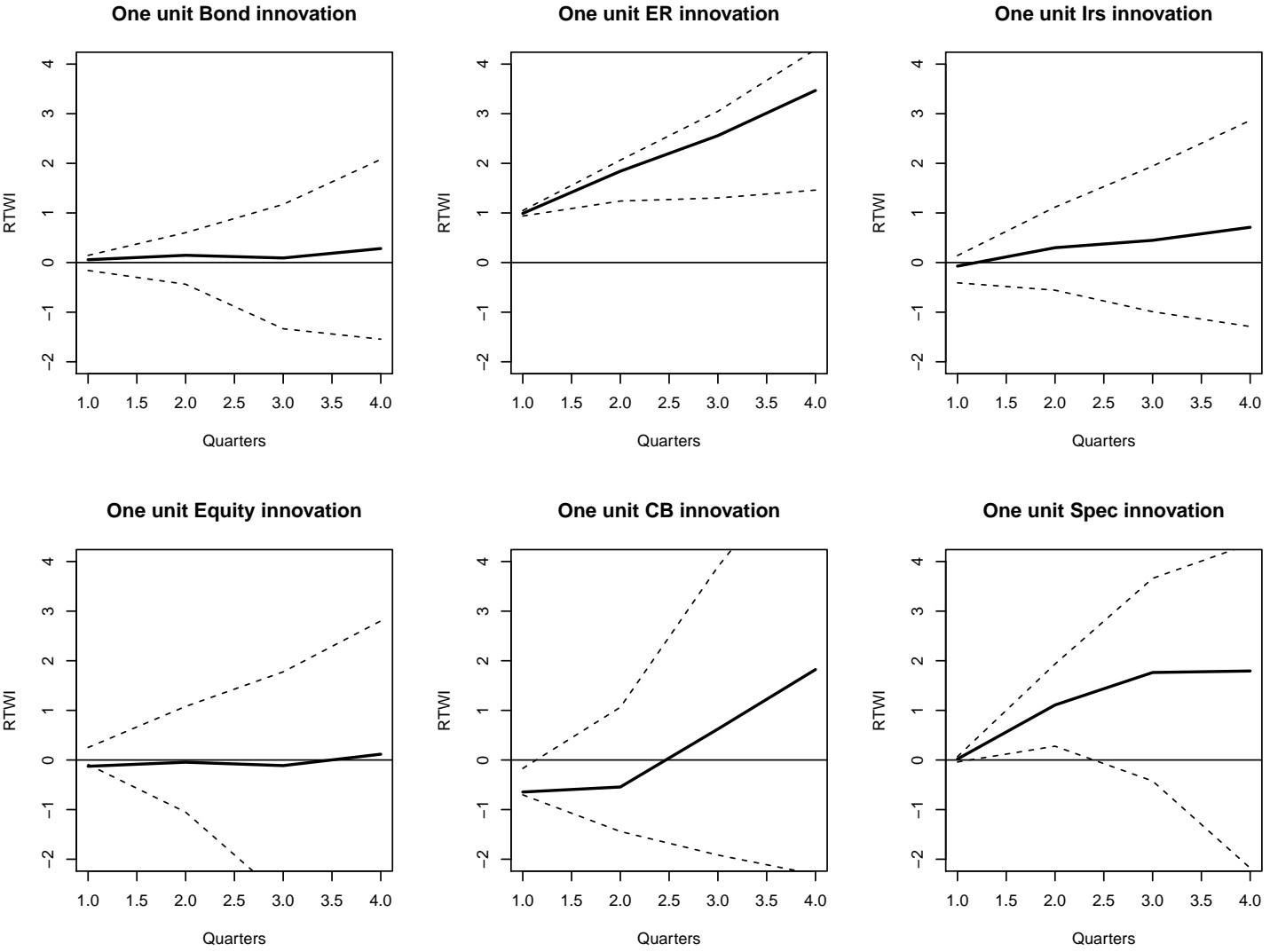

**Figure 3.** Impulse Response Functions for RTWI: Cholesky decomposition.

### 5. Conclusions

An SVAR model of the real exchange rate and international capital flows is estimated and IRF are used to analyse the effect of shocks to the system. Unlike most of the previous studies our model includes a role for short-term movements of capital that are linked to speculation about future exchange rate values and interest rate differentials. These can capture deliberate demand for foreign currency that require a price concession but that are not directly associated with equity and debt. Impulse Response Functions from the SVAR show that deviations from PPP can be explained by innovations in net international capital flows but, contrary to some of the other investigations of this issue, the type of flow that has the most pronounced and significant effect is that associated with speculation or momentum. Speculation associated with interest rate differentials as well as flows into equities, bonds and for FDI do not appear to be as important. In addition to the effect on competitiveness, these short-term capital flows can add liquidity to the banking system, fuel a credit book and cause an appreciation in asset prices.

The evidence presented here is consistent with the idea that speculative inflows and outflows of capital that are driven by sentiment can lead to significant changes in competitiveness and therefore they can influence economic growth, the price level and the range of policy options that are available. These findings add weight to the belief that policymakers should monitor these flows carefully to identify vulnerabilities in the international financial system. At the moment policy makers and the research community have concentrated on the gross or net flow of capital to equity or debt markets. For example, the IMF integrated approach to concentrates wholly on portfolio flows (Gelos et al. 2019).

This will require more comprehensive data. The current study uses a measure of speculation that is US-focused and based on the currencies of a small subset of developed market economies. This is certainly one limitation of the research. More generic or systematic measures of speculation would allow a more general test of the effect of these speculative flows on exchanges rates. What is required is a more granular view of the drivers of banking flows to identify those that are short-term and speculative from those that are accommodating deliberate decisions to buy and sell currency for other reasons. Hedge funds and other shadow banking institutions, for example, are more likely to drive the destabilising sudden-surges and sudden-stops that have led to calls for increased control over international capital flows. Therefore a better understanding of international financial speculation may allow policy-makers to fine tune capital controls to limit those that are more damaging while allowing those that can be beneficial through their development of capital and the sharing of risk.

**Author Contributions:** Conceptualization, R.H.; methodology, R.H.; software, R.H.; validation, R.H.; formal analysis, R.H.; investigation, R.H.; resources, R.H.; data curation, R.H.; writing—original draft preparation, R.H.; writing—review and editing, A.G.; visualization, R.H.; supervision, A.G.; project administration, R.H. All authors have read and agreed to the published version of the manuscript.

**Funding:** This research received no external funding.

**Institutional Review Board Statement:** Not applicable.

**Informed Consent Statement:** Not applicable.

**Data Availability Statement:** From the US Treasury: Transactions with Foreigners in Long-Term Securities. The interest rate data is constructed from data that is available from the IMF World Economic Database and the data for speculative positions is constructed from data that is available from the CFTC Committment of Traders Report. The data that was constructed from these public data sources is available on Github and the R code to run the SVAR model is available here and here. Accessed 22 April 2021.

**Acknowledgments:** I would like to acknowledge the help that has been provided by two anonymous referees.

**Conflicts of Interest:** The authors declare no conflict of interest.

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
