# Peer review of "International Capital Flows and Speculation"

_jrfm, doi:10.3390/jrfm14050197_

Round 1

Reviewer 1 Report

Thank you for the article, the topic of which certainly has potential, but the output in this form contains a number of major shortcomings, as well as minor shortcomings listed below. I recommend significantly reworking the article so that it can be acceptable for publication.

  1. The abstract is too short, it does not contain other important information. Please extend the abstract to a standard scope and provide the required information, which is also in the journal template (background, methods, results, conclusions).
  2. The citation "Claessens et al." on line 20 is missing a year.
  3. Part 1. Introduction is too short, it contains only a few references to previous literature. You need to add a detailed literature research with a sufficient number of quality and the latest possible sources. The sources used in the introduction are for the most part relatively old, certainly there are more recent research on the topic. The literature research is partly presented in Part 2. Materials and Methods in the first two paragraphs. This information must be part of the literature research, not the methodology.
  4. Abbreviations are not explained when first mentioned in the text.
  5. The name of table 1 is too long - the name should contain only a brief name of the table, while the description of the table should not be part of the name, but the text of the article, or notes below the table.
  6. In the Results section, you refer to Figure 1 and Figure 2, which you present after the text, ie in appendices that are not marked. Overall, this section is too brief.
  7. There is no "Conclusion" section at all, which should certainly be included in the scientific text. Only the Discussion part is presented, but without references to other literature - a comparison with the results of other authors.
  8. References at the end of the document are also formatted incorrectly. Each quote looks different.
  9. The overall arrangement of the article into individual parts is, in my opinion, inappropriate, although often interesting ideas and information are presented, but unstructured. Understanding all the contexts is so very difficult.
  10. The defined goal (aim) of the work is completely missing. The research article should also ideally contain research questions or hypotheses.

Author Response

Thank you for your comments.  They were extremely useful.  I have made the following changes in response to the questions that you raise.  

1) I have extended the abstract. 

2) Claessens year has been added. 

3) Literature review has been extended. 

4) I have checked the abbreviations

5) Table 1 name has been reduced. 

6) Results have been extended.  IRF have been incorporated into the text.

7) Conclusion has been added and extended. 

8) I have worked on the references to make the similar.

9) I have adjusted the text to a more conventional style.  I hope that this is more coherent. 

10) I have tried to highlight the importance of the work and the reason for the study.  

I look forward to any additional thought that you have. 

Regards, 

Rob Hayward

Reviewer 2 Report

This is an interesting paper that could be potentially publishable subject to some revisions that are discussed in more detail below.

Detailed comments:

Methods: Specify the SVAR model in text by providing the necessary equations.

Results: Confirm that there is no evidence of serial correlation in the residuals by means of relevant tests.

Discuss on the generalization of the results of the study.

Implications: Discuss on the theoretical and practical implications of the study.

Limitations: Discuss on the limitations of the study.

Minor comments:

The text needs some editing:

P2, line 43: Revise the sentence.

P2, line 82: Revise the sentence.

P3, line 110: Revise the sentence.

Author Response

Thank you for your comments.  They were extremely useful.  I have the following response. 

1) I  have provided equation for the SVAR.   These are on lines 198 to 219. 

2) I confirm that there is no evidence of serial correlation. Lines 277-280. 

3) I have extended the discussion of the results and the conclusion to cover the importance of finding data that can capture speculation and banking flows.  Lines 343-345. 

4) I have extended the practical implications of the study.  These are also in 333-343-345. 

5) I have added some limitations.   Lines 333-343. 

There is extensive re-writing. 

I look forward to your comments. 

Regards, 

Rob Hayward

Round 2

Reviewer 1 Report

Thank you for the changes, which helped a lot and the article is now publishable. I have only minor formal comments on Figure 2 and Figure 3, whose title is outside the page area. Another small comment is on the reference "Lane, P. and G. Milesi-Ferretti (2008)", where the names of the authors are given only by an abbreviation, while for other citation records they are listed in full (the following reference refers to the same authors and full names).